# *Phaeanthus vietnamensis* Ban Ameliorates Lower Airway Inflammation in Experimental Asthmatic Mouse Model via Nrf2/HO-1 and MAPK Signaling Pathway

**DOI:** 10.3390/antiox12061301

**Published:** 2023-06-19

**Authors:** Thi Van Nguyen, Chau Tuan Vo, Van Minh Vo, Cong Thuy Tram Nguyen, Thi My Pham, Chun Hua Piao, Yan Jing Fan, Ok Hee Chai, Thi Tho Bui

**Affiliations:** 1Department of Anatomy, Jeonbuk National University Medical School, Jeonju 54896, Jeonbuk, Republic of Korea; 2Faculty of Biology and Environmental Science, University of Science and Education, The University of Danang, Danang 59000, Vietnam; 3Institute for Medical Sciences, Jeonbuk National University, Jeonju 54896, Jeonbuk, Republic of Korea

**Keywords:** airway inflammation, asthma, oxidative stress, MAPK, *Phaeanthus vietnamensis*

## Abstract

Asthma is a chronic airway inflammatory disease listed as one of the top global health problems. *Phaeanthus vietnamensis* BÂN is a well-known medicinal plant in Vietnam with its anti-oxidant, anti-microbial, anti-inflammatory potential, and gastro-protective properties. However, there is no study about *P. vietnamensis* extract (PVE) on asthma disease. Here, an OVA-induced asthma mouse model was established to evaluate the anti-inflammatory and anti-asthmatic effects and possible mechanisms of PVE. BALB/c mice were sensitized by injecting 50 μg OVA into the peritoneal and challenged by nebulization with 5% OVA. Mice were orally administered various doses of PVE once daily (50, 100, 200 mg/kg) or dexamethasone (Dex; 2.5 mg/kg) or Saline 1 h before the OVA challenge. The cell infiltrated in the bronchoalveolar lavage fluid (BALF) was analyzed; levels of OVA-specific immunoglobulins in serum, cytokines, and transcription factors in the BALF were measured, and lung histopathology was evaluated. PVE, especially PVE 200mg/kg dose, could improve asthma exacerbation by balancing the Th1/Th2 ratio, reducing inflammatory cells in BALF, depressing serum anti-specific OVA IgE, anti-specific OVA IgG1, histamine levels, and retrieving lung histology. Moreover, the PVE treatment group significantly increased the expressions of antioxidant enzymes Nrf2 and HO-1 in the lung tissue and the level of those antioxidant enzymes in the BALF, decreasing the oxidative stress marker MDA level in the BALF, leading to the relieving the activation of MAPK signaling in asthmatic condition. The present study demonstrated that Phaeanthus vietnamensis BÂN, traditionally used in Vietnam as a medicinal plant, may be used as an efficacious agent for treating asthmatic disease.

## 1. Introduction

Asthma is a chronic airway disease that has been mentioned as a critical global health problem by the WHO organization. Asthma can develop in people of all ages and affected more than 262 million people worldwide in 2019, and its prevalence is rising [1]. Pathologically, asthmatic patients experience a thickening of their airway wall, mucus overproduction, and leukocyte infiltration [2]. Asthma is characterized by an imbalance in the T helper Th1/Th2 ratio and inflammatory cells, such as eosinophils and mast cells [3]. Th2 cells produce proinflammatory cytokine IL-4, IL-5, and IL-13 by activating GATA binding protein 3 (GATA3) in conjunction with signal transducer and activator of transcription 6 (STAT6) [4]. Th1 predominantly secretes interferon (IFN)-γ and interleukin (IL)-12 to inhibit Th2 cell differentiation by preventing STAT6 binding to the IL-4 receptor [5]. Reactive oxygen species (ROS) mediate oxidative stress and are responsible for airway inflammation. Although a powerful antioxidant system exists in the lung, the excessive production of ROS causes an imbalance of oxidants/antioxidants, leading to the harmful pathophysiological disorders associated with allergic asthma [6]. Generally, ROS is detrimental to the cell by lipid peroxidation of polyunsaturated fatty acids and oxidation of the DNA base and protein. Malondialdehyde (MDA) is a final stable product of lipid peroxidation and, thus, has been used as an oxidative stress marker [7]. An oxidative stress state also stimulates the activation of MAPK signaling, which plays a major regulatory role in pro-inflammatory cytokines secretion and the activation of the downstream signaling events that lead to inflammation [8]. The three major groups of MAP kinase include ERK1/2, JNK, and p38; each has been shown to stimulate, when in their phosphorylated form, the synthesis of pro-inflammatory cytokines and immune responses in the asthmatic mine [9]. Nuclear factor erythroid 2-related factor 2 (Nrf2)/heme oxygenase-1 (HO-1) is also a major system that regulates the work of antioxidant proteins and can help protect against oxidative damage [10]. Several recent studies have confirmed that Nrf2/HO-1 signaling is one of the most promising targets for treating asthma [11,12]. Mice lacking an Nrf2 gene experience serious allergen-driven airway inflammation and bronchoconstriction [12].

Inhaled corticosteroids are considered the most effective medical treatment available for asthma [13]. It is only effective in treating symptoms by controlling inflammatory processes. While corticosteroids are effective, their use is often associated with undesirable adverse effects, especially at high doses, such as a sore mouth or throat, oral thrush, and irregular or fast heartbeat [14]. However, prolonged or high-dose use of corticosteroids can lead to steroid resistance, resulting in a decreased therapeutic response, requiring higher doses or alternative treatment strategies to achieve the desired effects [15]. *Phaeanthus vietnamensis* Ban is a well-known medicinal plant in Vietnam with anti-oxidant, anti-microbial, and anti-inflammatory potential, as well as gastro-protective properties [16,17,18]. Traditionally, *P. vietnamensis* Ban has been used for several inflammatory diseases, such as sore red eyes, diarrhea, and pustule [16]. *P. vietnamensis* Ban ethanol 70% extract (PVE) is rich in spathulenol, neophytadiene, octadecanoic acid ethyl ester, n-hexadecanoic acid, oleic acid, linoleic acid ethyl ester, and stigmasterol, which have an excellent anti-allergic and anti-inflammatory properties [19,20,21,22,23]. Neophytadiene has demonstrated beneficial effects, such as pain relief, fever reduction, anti-inflammatory action, and antioxidant properties. N-hexadecanoic acid has been used for the treatment of rheumatic symptoms by inhibiting the phospholipase A_2_ effect [23]. Much evidence has suggested the beneficial effects of oleic acid (OA) in cancer, autoimmune, and inflammatory diseases, besides its ability to facilitate wound healing [24]. In the European Union, stigmasterol is listed as a food additive that can improve low-density lipoprotein cholesterol (LDL-cholesterol) levels and neuroprotective activities [25]. However, the efficacy of PVE in treating asthma remains uncertain. For these reasons, this study investigated whether PVE was an effective treatment in the ovalbumin (OVA)-induced allergic asthma mouse model.

## 2. Materials and Methods

### 2.1. Phaeanthus vietnamensis Extract Preparation

*Phaeanthus vietnamensis* was collected directly from a mountain (Danang, Vietnam). The voucher specimens of *P. vietnamensis* DND 256 were stored in the Herbarium of the Botany Museum, Vinh University, Vietnam. Raw *P. vietnamensis* material was extracted with 10× volumes of 70% ethanol under heating conditions (65 ± 5 °C) for 8 h, and the extract solution was collected using vacuum filtration (Filtstar, SLOSF03001, Shenzhen, China). This process was repeated three times. The filtrate was then concentrated with a rotary evaporator (IKA, L_IKA_31035, Staufen, Germany) and dried in a freeze-dryer (FD4, Pathumthani, Thailand) to obtain the extract powder. The extraction efficiency reached 3.1%. The powders were kept at 4 °C. PVE powder was dissolved in saline before further experiments.

### 2.2. Chemical Constituents PVE Analyzed through UPLC-Q-TOF-MS/MS

The chemical constituents of PVE were analyzed using UPLC-Q-TOF-MS/MS. In total, 3 μL of 4 mg/mL of PVE was loaded into an ACQUITY UPLC HSS C18 column (100 mm × 2.1 mm, 1.8 μm) with the mobile phase of 0.1% formic acid (solution A) and acetonitrile (solution B) for gradient elution (0–8 min, 1–5% B; 9–21 min, 5–20% B; 21–24 min, 20–25% B; 24–30 min 25–40% B; 30–36 min, 40–75% B; 36–42 min, 75–90% B; 42–45 min 99% B). An amount of 3 μL of the sample was injected, and the flow rate was 0.25 mL/min. The optimal MS parameters were ion source gas1: 45, ion source gas2: 55, curtain gas: 35, source temperature: 600 °C, the ion spray voltage floating: 5500 V/−4500 V, TOF MS scan *m*/*z* range: 100–1500 Da, product ion scan *m*/*z* range: 25–1500 Da, TOF MS scan accumulation time: 0.2 s/spectra, and product ion scan accumulation time: 0.03 s/spectra. Sodium formate was used for the purpose of TOF analyzer calibration. The data were analyzed using SCIEX OS software version 3.0. The ESI-MS negative molecular ion peaks, MS/MS ions, and retention time, along with the references and TCM MS/MS Library, were used to tentatively identify the chemical constituents.

### 2.3. Animal Protocol

Male five-week-old BALB/c mice were acquired from Damool Science (Dae-jeon, Korea). The mice were acclimated to laboratory conditions (23 ± 3 °C humidity, about 50 ± 10%, and 12-h light/dark cycles) for 1 week prior to the commencement of the experiments. All animal experiments were performed at the Department of Anatomy, Medical School at Jeonbuk National University and were approved by the Institutional Animal Care and Use Committee (JBNU 2021-0115).

The asthma mouse model was established following our previous study [26] with small modifications. Briefly, the mice were first randomly divided into six groups, 6 mice in one group: (1) Naive; (2) OVA; (3) PVE 50; (4) PVE 100; (5) PVE 200; and (6) Dex. The animal protocols are briefly described in Figure 1. OVA-induced asthma mouse models were sensitized on the first day, and on day 15, they received an intraperitoneal injection of 200 µL of 50 μg OVA (GradeV, Sigma-Aldrich, St. Louis, MO, USA) and 1 mg aluminum-containing adjuvants (Thermo Scientific, Rockford, MD, USA). After sensitization, the mice were challenged with 20 mL OVA (5%, *w*/*v*) for 20 min by aerosol on days 27, 28, and 29. Between days 15 and 29, mice in the PVE 50, PVE 100, and PVE 200 groups received a once-daily oral treatment with PVE at doses of either 50, 100, or 200 mg/kg, while mice in the Dex group were orally administrated Dex 2.5 mg/kg. Mice in the OVA groups were given saline. Mice in the Native group did not get any sensitization, treatment, or challenge. All mice were sacrificed 24 h after the last OVA challenge. The animal experiment was performed 3 times individually.

### 2.4. Cytotoxicity MTT Assay

An MTT colorimetric assay was performed using an MTT assay kit (ab211091, Abcam, Cambridge, UK) to test the cell toxicity of the PVE on rat peritoneal mast cells (RPMC). RPMCs were purified by centrifugation on a medium containing Percoll (Sigma-Aldrich, St. Louis, MO, USA). RPMCs with purity 95–98% (2 × 10^5^ cells/well) were pretreated with PVE (0.01, 0.1, and 1 mg/mL) at 37 °C for 3 h. Cells were incubated in MTT (1000 mg/mL) at 37 °C for 1 h. Finally, the absorbance was measured at 570 nm with an absorbance microplate reader.

### 2.5. Rat Peritoneal Mast Cells (RPMCs) Degranulation Assay

RPMCs were confined as previously described elsewhere [27]. Rat was euthanized, and death was confirmed with the loss of reflexes. An amount of 10 mL of ice-cold RPMI medium was injected into the peritoneal cavity using a 10 mL syringe equipped with a 27 G needle. The rat abdomen was massaged for about 2 min to detach peritoneal cells into the RPMI medium. The rat abdomen was opened to gently and slowly aspirate the fluid from the peritoneal cavity into a 50 mL tube. Pour 40 mL RPMI medium into the cavity to separate the remaining mast cells. Again, aspirate the fluid into the previous 50 mL tube. The collected cells solution was centrifuged at 1000 rpm, 4 °C, for 10 min. Discard the supernatant and wash the cells with HEPES twice. The mast cells were purified (about 95%) with Percol gradient. Purified RPMCs were resuspended in HEPES pH 7.4 (Cat. 15630106, Thermo Fisher, Rockford, MD, USA) buffer. RPMCs were pretreated with PVE (1, 0.1, 0.01 mg/mL) or saline for approximately 10 min at 37 °C, then incubated with C48/80 at 5 µg/mL (C2313-250, Sigma-Aldrich, St. Louis, MO, USA) or saline for about 15 min. The percentage (%) of degranulated mast cells and the morphology of the RPMCs were then observed under a microscope.

### 2.6. Bronchoalveolar Lavage Fluid Analysis

Bronchoalveolar lavage fluid (BALF) was collected by pushing 1 mL of cold saline into the trachea and withdrawing it through the cannula. BALF was centrifuged (at 10,000 rpm, 4 °C, 10 min); then, the supernatant was stored at −70 °C for further experiments. The pellets were dispersed in 1 mL saline. Total cell numbers were counted in a hemocytometer.

To examine differential cell numbers, the saline solutions that contained cells were centrifuged onto slides with a Cytospin device (Centrifuge 5403, Eppendorf, Hamburg, Germany). Cells were stained with a Diff-Quik Staining reagent to quantify the inflammatory cells with a light microscope (Leica, Teaneck, NJ, USA) at ×400 magnification [28].

### 2.7. Histopathology and Immunohistochemistry

The lung of the mice (which were not taken the BALF) were used to check histopathology. They were fixed in neutral buffered formalin 10% dehydrated with a gradually increased concentration of alcohol and xylene. Lung tissues were embedded in paraffin and then sectioned (4 µm thick) for several kinds of stain, hematoxylin and eosin (H&E) (Sigma-Aldrich, St. Louis, MO, USA) to study lung structure, periodic-acid–Schiff (PAS) (ab150680, Abcam, Cambridge, UK) to study inflammatory cell infiltration, and Masson’s Trichrome (25088, Polysciences Inc, Warrington, PA, USA) to research mucus production and collagen deposition.

The lung tissues were also stained via an immunohistochemistry method using an α-SMA antibody (ab124964, Abcam, Cambridge, UK) at 1/400 to confirm the fibrosis feature. Perform heat-mediated antigen retrieval with citrate buffer pH 6 (ab93678, Abcam, Cambridge, UK) before commencing with the IHC staining protocol. The IHC process was performed using Rabbit specific HRP/DAB (ABC) Detection IHC kit (ab64261, Abcam, Cambridge, UK) according to the manufacturer’s instructions. The sections were counterstained with Harris hematoxylin (MFCD00078111, Sigma-Aldrich, St. Louis, MO, USA). The lung inflammation core was evaluated, following the scoring criteria from Baris et al. [29]. The goblet cell, collagen fiber, and fibrosis-positive areas were measured with Fiji software version 2.9.0.

### 2.8. Measurement of Immunoglobulins and Cytokine Using ELISA

The levels of anti-OVA specific IgE (439807, Biolegen CNS Inc., San Diego, CA, USA), anti-OVA specific IgG_1_ (500830, Cayman Chemical, Ann Arbor, MI, USA), which are related to Th2 cells, anti-OVA specific IgG_2a_ (3015, Chondrex Inc., Woodinville, WA, USA), which is related to Th1 cells, histamine (ab213975, Abcam, Cambridge, UK) in the serum, IFN-γ and IL-12, IL-4, IL-5, IL-13 (R&D Systems Inc., Minneapolis, MN, USA), GATA3, and Eoxtain (Abcam, Cambridge, UK) in BALF were quantified using ELISA kits consistent with the manufacturers’ instructions.

### 2.9. Western Blot

The presence of Th1-transcription factor T-bet, Th2-transcription factor GATA-3, antioxidant protein HO-1, MAPK signaling-related proteins c-JNK, ERK, p-38, and their phosphorylated form was determined via Western blot. Lung tissues were homogenized with RIPA buffer (EBR001-1000, Enzynomic, Daejeon, Republic of Korea) to extract proteins, then quantified by BSA standard with Bradford dye (Bio-Rad Laboratories, Inc., Hercules, CA, USA). The samples were loaded onto an SDS-PAGE 10% and then transferred onto an activated polyvinylidene difluoride (PVDF) membrane (Bio-Rad Laboratories, Inc., Hercules, CA, USA). The membrane was blocked with 5% skimmed milk for approximately 1 h, then incubated with diluted primary antibody at 1/1000 BSA 5% overnight at 4 °C in an orbital shaker. After washing off the primary antibody with TBST, the membrane was then incubated with a secondary antibody for about 1 h at room temperature. The blot was detected using an ECL substrate. The density of bands from the membrane was measured using ImageJ software version 13.0.6.

### 2.10. Statistical Analysis

All results were obtained using Graph Pad Prism software (v5.0, La Jolla, CA, USA). All data were calculated as means SEM. The statistical differences among the groups were analyzed with one-way ANOVAs followed by Tukey’s test and considered at *p* < 0.05.

## 3. Results

### 3.1. Analyzed Compounds from Phaeanthus vietnamensis

This study identified a total of 16 compounds that separated well and were identified from the ethanol extract of *P. vietnamensis* with UPLC-Q-TOF-MS/MS, which is shown in Figure 2. The main important biomolecules identified were spathulenol, neophytadiene, octadecanoic acid ethyl ester, n-Hexadecanoic acid, hexadecanoic acid ethyl ester, oleic acid, linoleic acid ethyl ester, stigmasterol, neophytadiene, and octadecanoic acid ethyl ester. These biomolecules are known to possess anti-inflammatory, antioxidant, and antimicrobial effects [19,20,22]. Oleic acid, linoleic acid, and stigmasterol have been reported to have antioxidant, anti-inflammatory, anti-allergic, and antihistaminic effects [21,23,30]. Therefore, *P. vietnamensis* extract is thought to contribute to inflammation related to its important biomolecules present.

### 3.2. PVE Is Non-Toxic to the Rat Peritoneal Mast Cell (RPMC)

MTT assay was performed to determine the dose of PVE that may have cytotoxicity to the RPMC. As shown in Figure 3A, the viable cells were almost 100% after 3 h of treatment with PVE at doses of 0.01, 0.1, and 1 mg/mL PVE. However, the dose of 10 and 100 mg/mL of PVE sharply decreased the cell viability to lower than 50% when compared with the control group. Therefore, PVE at doses of 0.01, 0.1, and 1 mg/mL were used in the next experiment.

### 3.3. PVE Prevented Degranulation of RPMCs by Compound 48/80

To examine the protective effect of PVE on mast cell-mediated allergic responses, RPMC pretreated with deferent doses of PVE was then incubated with compound C48/80. Figure 3D showed that RPMCs were spherical or oval and included numerous secretory granules. The morphology of RPMCs set with PVE (1, 0.1, 0.01 mg/mL) was similar to that of normal RPMCs. However, after incubation with compound 48/80 for 15 min, the membranes of RPMCs become thinner or perforated, and cells showed extensive degranulation. They had burst to release multiple granules from the cell surface. We observed a decrease in mast cell degranulation in PVE-treated groups, especially at the high-dose PVE. Furthermore, the level of histamine released from degranulated mast cells was higher in the C48/80 group when compared with the Naïve group. However, the level of histamine released to the supernatant was significantly decreased in the pre-incubating with PVE 1 mg/mL group when compared with the positive control group (Figure 3C). Figure 3B reflects that, in C48/80-treated RPMCs group, the percentage of degranulated mast cells had significantly increased compared to the control group. However, PVE treatment with other doses (1, 0.1, 0.01 mg/mL) significantly suppressed mast cell degranulation.

### 3.4. PVE Reduced the Recruitment of Inflammatory Cells in BALF of the OVA-Induced Asthmatic Mouse Model

To examine whether PVE could suppress airway inflammation, the number of inflammatory cells infiltrated in BALF was quantified. Mice in the OVA group showed elevated levels of infiltration and the accumulation of eosinophils, neutrophils, and macrophages in the lung compared to the Naive group, leading to an increase in the total cell number. In contrast, PVE treatment (100 and 200 mg/kg) significantly suppressed the increase in eosinophils and total cells as compared to OVA-induced mice. In addition, we performed a Diff-Quick stain to classify the presence of differential inflammation cells in BALF. In Figure 4A, red circles indicate eosinophils, and yellow circles mark neutrophil infiltration. As this figure highlights, the number of eosinophils is clearly lower in the BALF of PVE-and Dex-treated mice (Figure 4B).

### 3.5. PVE Reduced Asthma Histopathological Features in Lung Tissue of the OVA-Induced Asthmatic Mouse Model

The bronchus histopathological changes in the lung tissue were assessed via the H&E staining. The inflammation score was based on the scoring criteria from Baris et al. [29]. The bronchus showed significant changes for mice in the OVA group compared to Naive mice. Specifically, the airway epithelium swelled, the smooth muscle layer was thickened, and there were a number of inflammatory cells surrounding the bronchial that had been penetrated. However, inflammatory features were alleviated in the PVE (100 and 200 mg/kg) treatment groups (Figure 5A,B). PAS staining showed mucus hypersecretion as a violet color in the bronchial airways (black arrow). The administration of Dex and PVE (100 and 200 mg/kg) suppressed mucus overproduction in the bronchial epithelium (Figure 5A,C).

Masson’s trichrome staining revealed that PVE-treated mice (100 and 200 mg/kg) had collagen proliferation edged around the peribranchial and perivascular less when compared with the asthma mice. Collagen deposition was observed as a blue color in the peribranchial and perivascular (Figure 5A,D). The IHC stain with α-SMA marker results also showed the same trend as Masson’s trichrome staining. The fibrosis state in the lung tissue of asthma mice appeared more severe than in the naïve group. With the treatment with PVE at the high dose of 200 mg/kg or Dex, the fibrosis condition was alleviated (Figure 5A,E)

### 3.6. PVE Reduced The Levels of OVA-Specific Antibodies and Histamine in the Serum of the OVA-Induced Asthmatic Mouse Model

The levels of anti-OVA specific IgE, anti-OVA specific IgG_1_, and histamine were significantly more up-regulated in the OVA group than those in the Naive group. However, PVE treatments (100 and 200 mg/kg) decreased anti-OVA-specific IgE, anti-OVA-specific IgG_1_, and histamine (compared with those in the OVA-induced mice). In contrast, lowered anti-OVA-specific IgG_2a_ levels were observed in the OVA group, while higher levels were seen in the PVE groups (Figure 6A–D).

### 3.7. PVE Restored the Balance of Helper T Cell Responses in the BALF of the OVA-Induced Asthmatic Mouse Model

We evaluated the levels of Th1-related cytokines (IFN-γ, IL-12) and Th2-related cytokines (IL-4, IL-5, IL-13, eotaxin) in BALF to determine whether PVE regulated the balance of helper T cell responses. In the OVA group, IL-4, IL-5, IL-13, and eotaxin levels were significantly higher than those observed in the Naïve group. In contrast, PVE oral administrations (100 and 200 mg/kg) resulted in a marked decrease, compared with the OVA-induced mice, in the levels of IL-4, IL-5, IL-13, and eotaxin (Figure 7A–D). Our experiments resulted in a decrease in IL-12 level in the OVA group compared to the Naïve group and a significantly elevated IL-12 level in PNE-treated mice (100 and 200 mg/kg) compared with the OVA-induced mice (Figure 7E). Meanwhile, no significant differences in the levels of IFN-γ were observed between the PVE and OVA groups (Figure 7F). The Th1-transcription factor T-bet expression in the lung tissue of OVA mice was suppressed, while the Th2-transcription factor GATA-3 expression was increased when compared with the Naïve group. Interestingly, the level of T-bet was significantly enhanced with the PVE treatment. Furthermore, the expression of GATA-3 was remarkably decreased in the PVE or Dex treatment group (Figure 8A–C). The T-bet/GATA ratio, which represents the Th1/Th2 balance, was notably improved with the treatment of PVE in a dose-dependent manner (Figure 8D). These results suggest that the oral administration of PVE may regulate the balance of Th1/Th2 responses by enhancing the Th1 and suppressing the Th2 response.

### 3.8. Effect of PVE on Oxidative Stress in the OVA-Induced Asthmatic Mouse Model

To determine the mechanism by which the PVE’s antioxidant affects asthma, the marker of oxidative stress MDA and Nrf2/HO-1 signaling pathway-related protein were determined with ELISA and Western blot. Figure 9 highlights that in the asthmatic mice, the level of MDA had sharply increased when compared to Naïve mice. However, after PVE administration, the level of MDA remarkably decreased. The oxidant/antioxidant balance was restored by up-regulating antioxidant proteins Nrf2 and HO-1 and down-regulating oxidant enzyme MDA by PVE.

### 3.9. PVE Inactivates the MAPK Signaling Pathway in the OVA-Induced Asthmatic Mouse Model

Oxidative stress may activate some important signaling pathways that amplify inflammation, for example, the Mitogen-activated protein kinases (MAPKs) pathway. We evaluated the expression of the MAPK signaling-related proteins, including JNK, p38, and ERK, as well as their phosphorylated forms p-JNK, p-p38, and p-ERK with Western blot. As Figure 10 shows, the expression of phosphorylated form p-p38 and p-JNK had clearly increased in the asthmatic mice, but this expression was effectively decreased by treatment with PVE or Dex.

## 4. Discussion

PVE has long been harvested from the wild for use as a medical treatment for some inflammatory diseases, non-specific conjunctivitis, gastrointestinal disorders, and for disinfection of wounds. PVE is rich in spathulenol, oleic acid, and several bioactive compounds, including neophytadiene, n-hexadecanoic acid, and n-methylcorydaldine. In previous studies, spathulenol and oleic acid are reported to have anti-oxidant, anti-inflammatory, anti-proliferative, and antimycobacterial properties [20,31]. Prior to this work, however, no study has examined the effect of PVE on asthma. An OVA-induced asthma mouse model was established to evaluate the anti-inflammatory and anti-asthmatic effects and possible mechanisms of PVE. Our results demonstrated that the administration of PVE, particularly at a high dose, effectively mitigated asthma symptoms by restoring the balance between Th1 and Th2 immune responses. This was evidenced by a reduction in the number of inflammatory cells in bronchoalveolar lavage fluid (BALF) and the restoration of normal lung histology. We also found that in the PVE treatment group, the expression of antioxidant enzymes Nrf2 and HO-1 had significantly increased in lung tissue as well as the BALF and that treatment decreased the level of oxidative stress marker MDA in the BALF and inhibited the activation of MAPK signaling that is a feature of asthma.

Airways tend to be irritated by the inhalation of stimulating substances, which may provoke allergic reactions. OVA is a protein component in egg whites, commonly used to mimic allergy-induced asthma due to its association with the Th2 immune response [32]. In mice exposed to OVA, the differentiation of Naïve T cells toward Th2 cells disrupts the Th1/Th2 balance, leading to the development of asthma. [33]. More specifically, once the epithelial barrier exposure to allergens, the allergens can be sampled by dendritic cells (DCs). DCs mature and migrate to regional lymph nodes to contact naive T cells. In the presence of early IL-4, these naive T cells acquire the characteristics of Th2 cells, which produce more IL-4 and IL-13 and are essential for stimulating B cells to generate IgE. Allergen-specific IgE attaches to the IgE receptors (FcεRI) on tissue-resident mast cells to sensitize them to respond to the allergen in the next re-exposer [34]. IgG_1_ and IgG_2a_ immunoglobulin isotypes are known to be the two markers representative of Th1 and Th2 cells [35]. The Th2 cell cytokines (IL-5) recruit eosinophils and activate them; IL-13 is secreted to induce goblet cell hyperplasia and mucous hypersecretion in airway epithelial cells [36], causing the typical histological features of asthma, including epithelial shedding, goblet cell hyperplasia, subepithelial fibrosis, thickened airway, smooth muscle mass, and inflammatory cell infiltration [37]. In asthmatic mice, due to chronic airway inflammation, the epithelial barrier was broken, leading to epithelial cell shedding. The activation of mast cells may release their cytokines and chemokines to recruit leukocytes and activate them. Consequently, we counted cell type percentages in BALF after staining with a Diff-Quick solution. In the asthmatic mice group, the number of epithelial cells and eosinophils was significantly increased. H&E, PAS, and Trichrome stain revealed that the lung tissue of mice in the OVA group also appeared to possess fully asthmatic features, including thickened airways, smooth muscle mass, goblet cell hyperplasia, and excess collagen fiber deposition. Moreover, the levels of IgE, IgG_1_, and histamine in the serum were dramatically up-regulated in the OVA mice group. In sum, there was ample evidence that the asthmatic mouse model had been successfully established. Oral PVE administration alleviated the inflammatory features, reduced mucus secretion, and lowered the number of collagen fibers in lung tissues. It also clearly down-regulated the level of IgE and histamine in the serum. It may suggest that PVE exerted an anti-asthmatic effect on the OVA-induced asthmatic mouse model.

Inflammation and oxidative stress are inextricably interrelated in airway allergic inflammatory diseases [38]. Asthma is well-known to be associated with both increased oxidant force and decreased cellular antioxidant capacity [39]. Oxidant generation is one necessary component of the metabolism that maintains cell homeostasis across many types of cells [40]. The inhalation of allergens and pollutants promotes the production of reactive oxygen species leading to lipid peroxidation, airway inflammation, hyper-responsiveness, and remodeling in the allergic asthma [39]. The accretion of inflammatory cells in the airway also speeds up the secretion of reactive oxygen species and suppresses antioxidative processes [41]. The inflammatory cell eosinophils, neutrophils, monocytes, and macrophages recruited to the asthmatic airways, as well as the resident cells (such as bronchial epithelial cells), possess an exceptional capacity for oxidant production in response to various stimuli [38]. MDA levels are often elevated in asthma, indicating an increased lipid peroxidation that is believed to contribute to the pathophysiology of the asthma [42]. To protect itself from oxidative stress, the lung has a well-developed antioxidant system, typically an Nrf2/HO-1 pathway. The multifunctional regulator of the Nrf2 has been identified as a cytoprotective factor that regulates the expression of genes that encode antioxidative enzymes or detoxifying molecules [43]. One of the genes regulated through Nrf2 is HO-1. This cytoprotective enzyme catalyzes heme degradation to equimolar amounts of iron ions, biliverdin, and CO, which are rapidly converted into bilirubin. HO-1 is, therefore, considered an impressive antioxidant that regulates several potent biological processes, including anti-inflammation, anti-apoptosis, anti-cell proliferation, and protection against the angiogenesis [44]. In this study, MDA levels were significantly decreased in the mice treated with high doses of PVE and Dex compared to the OVA group. Moreover, the levels of Nrf2 and HO-1 were notably improved by high doses of PVE compared to the OVA group. Accordingly, we infer that PVE exerted an anti-oxidative effect on the OVA-induced asthmatic mouse models.

Oxidative stress causes a reverse sequence of events, i.e., activating some kinases or signaling pathways, such as the MAPK signaling [45]. The MAPK signaling cascade consists of a family of protein kinases divided into three major groups: ERK; p38 MAPK; and JNK. Once MAPK is activated, it can be phosphorylated, translocated to the nucleus, and then bound to the targeting transcription factors to trigger the generation of pro-inflammatory cytokines and chemokines [46]. MAPK cascades are heavily involved in T cell activation, differentiation, proliferation, and cytokine production. P38 MAPK may promote the initial commitment of Naïve T helper cells toward the Th2 phenotype by mediating phosphorylation to facilitate the GATA-3 nuclear import [47]. Pyridinyl imidazole, an inhibitor of the p38 mitogen, has been reported to suppress the production of IL-4 and IL-5 by inhibiting CD28 [48]. JNK, also known as stress-activated protein kinases (SAPK), can be activated in response to the stress [49]. JNK and p38 also have been reported to involve in the expression of pro-inflammatory mediators [50]. MAPK also plays an important role in the activation and function of airway smooth muscle cells and epithelial cells during inflammation [51]. In this study, the administration of PVE improved the anti-oxidant Nrf2/HO-1 system, significantly lowering MDA levels and consequently deactivating the MAPK by suppressing the phosphorylation of JNK and p38. Therefore, the production of IL-4, IL-5, and GATA-3 was suppressed as a corollary. Taken together, we suggest that PVE produces an anti-oxidative via enhancing the Nrf2/HO-1 signaling pathway, and PVE exerts its anti-inflammatory effect via suppressing the MAPK signaling pathway in an OVA-induced asthmatic mouse model.

## 5. Conclusions

We demonstrated that administration of PVE by oral gavage meaningfully attenuated airway inflammation in OVA-induced AR mouse models by suppressing inflammation cells (eosinophils, goblet cells), reducing mucus hypersecretion and collagen deposition, regulating the balance of Th1 and Th2 cytokines, depressing serum anti-OVA IgE, anti-OVA IgG1, and histamine levels, and consequently, encouraging the anti-oxidative Nrf2/HO-1 system to inhibit the MAPK signaling pathway. Accordingly, we conclude that PVE exerts a significant anti-inflammatory effect on asthma and suggests that PVE may become an efficacious complementary agent in asthma prevention and treatment.

## Figures and Tables

**Figure 1 antioxidants-12-01301-f001:**
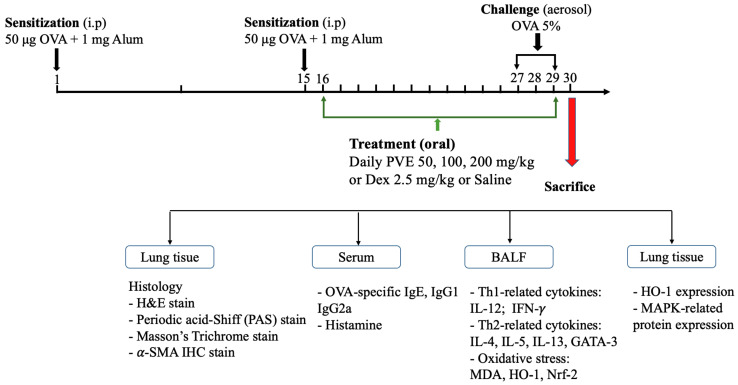
Animal experimental protocol. The male 5-week-old BALB/c mice were divided into 6 groups: group (1) Naïve; (2) OVA; (3) PVE 50; (4) PVE 100; (5) PVE 200; and (6) Dex. The asthma mouse model was established by sensitization OVA (i.p) on day 1 and day 15; then, the OVA challenge (nebulization) was performed from day 27 to day 29. The asthma mice in groups (3), (4), and (5) were orally administered with corresponding concentrations of PVE 50, 100, 200 mg/kg. The mice in the Dex group were treated with Dex 2.5 mg/kg. The mice in the OVA group were given Saline. All the mice were sacrificed on day 30. PVE = *Phaeanthus vietnamensis* extract; Dex = Dexamethasone; OVA = Ovalbumin.

**Figure 2 antioxidants-12-01301-f002:**
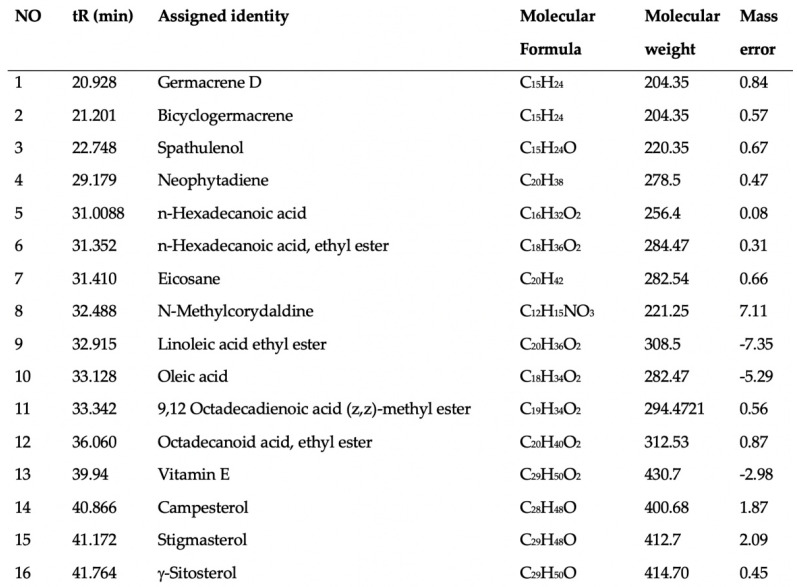
Analyzed compounds from Phaeanthus vietnamensis 70% ethanol extract with UPLC-Q-TOF-MS/MS. The main important biomolecules identified were spathulenol, neophytadiene, octadecanoic acid ethyl ester, n-hexadecanoic acid, hexadecanoic acid ethyl ester, oleic acid, linoleic acid ethyl ester, stigmasterol.

**Figure 3 antioxidants-12-01301-f003:**
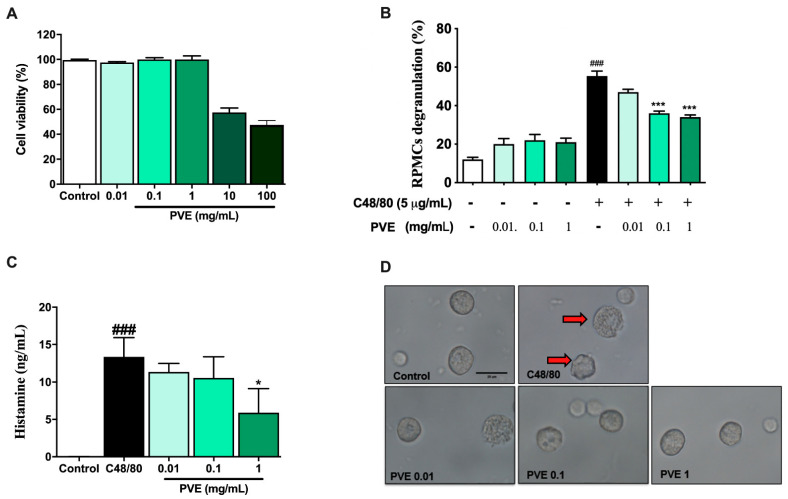
PVE prevented the degranulation of RPMCs by compound 48/80. (**A**) MTT assay. (**B**) PVE protected RPMCs degranulation from C48/80. (**C**) The level of histamine released from RPMCs. (**D**) Inverted light microscopy of RPMCs. RPMCs (2 × 10^5^ cells/well) were incubated with different concentrations of PVE (0.01, 0.1, and 1 mg/mL) at 37 °C for 3 h; then, absorbance was measured at 570 nm with a spectrophotometer. RPMCs were pretreated with PVE (10, 1, 0.1 mg/mL) or saline for 10 min at 37 °C and then incubated with C48/80 (5 µg/mL) or saline for 15 min. PVE dose-dependently inhibited the C48/80-induced RPMCs degranulation. The red arrow indicates degranulated mast cells. * compared to OVA; ###, *** *p* < 0.00, * *p* < 0.05. Scale bar = 25 µm. PVE = *Phaeanthus vietnamensis* extract; RPMC = Rat peritoneal mast cell; C48/80 = Compound 48/80.

**Figure 4 antioxidants-12-01301-f004:**
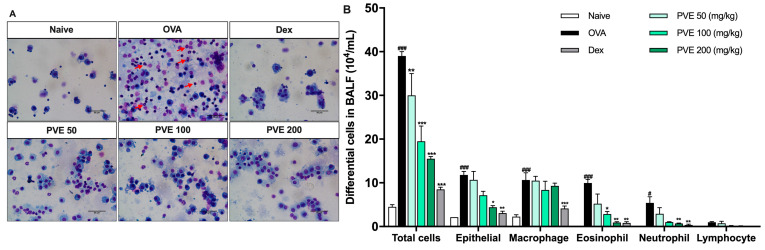
PVE reduced the recruitment of inflammatory cells in BALF of the OVA-induced asthmatic mouse model. (**A**) Diff-Quick stain. (**B**) The differential cells in BALF. The number of epithelial cells and inflammatory cells (macrophage, eosinophil, neutrophil) was significantly increased in BALF of the OVA group compared with the Naive group, and those cell numbers were significantly decreased in PVE 50, 100, 200 mg/kg, and Dex 2.5 mg/kg groups. The red arrows indicated eosinophils. # Compared to Naive; * compared to OVA. ###, *** *p* < 0.001; ** *p* < 0.01; #, * *p* < 0.05. BALF = Bronchoalveolar lavage fluid; Dex = Dexamethasone; PVE = Phaeanthus vietnamensis extract; OVA = Ovalbumin. Scale bar = 50 µm.

**Figure 5 antioxidants-12-01301-f005:**
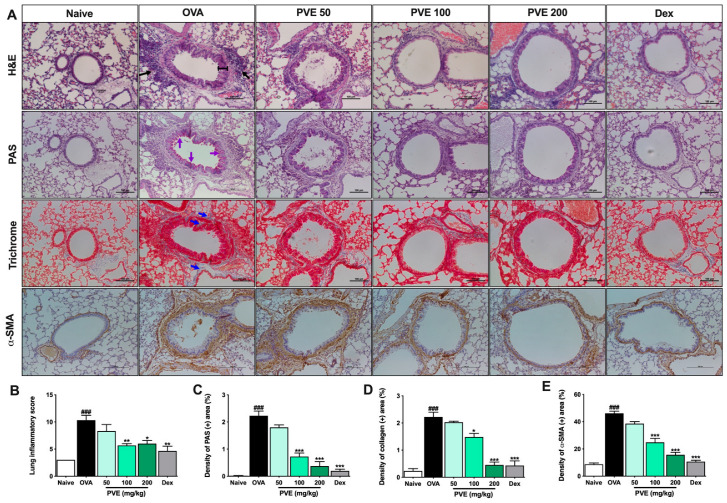
PVE prevented histopathological changes in the lung tissue of the OVA-induced asthmatic mouse model. (**A**) Histology of lung tissue. (**B)** Lung inflammatory score. (**C**) The density of PAS (+) area (%). (**D**) The density of collagen (+) area. (**E**) The density of α-SMA (+) area. All pictures were at a magnification of ×200. Via H&E staining, typical inflammation features were observed in lung tissue of asthma mice. Via PAS staining, asthma mice showed an increase in goblet cells resulting in oversecreted mucus into the lumen of the bronchia (which was stained with purple color and indicated by purple arrows). Via Trichrome staining, the collagen fiber (which was stained with blue color and indicated by blue arrows) was abundantly expressed surrounding the bronchi and vessels in the lung tissue of asthma mice. IHC stain with α-SMA again confirmed the majored fibrosis state in the OVA group. However, the inflammation feature, goblet cell hyperplasia, and collagen fiber deposition in asthma mice were considerably attenuated with PVE. * compared to OVA. ###, *** *p* < 0.001; ** *p* < 0.01; * *p* < 0.05. Dex = Dexamethasone; H&E = hematoxylin and eosin; IHC = Immunohistochemistry; PAS = Periodic acid–Schiff; PVE = *Phaeanthus vietnamensis* extract; OVA = Ovalbumin.

**Figure 6 antioxidants-12-01301-f006:**
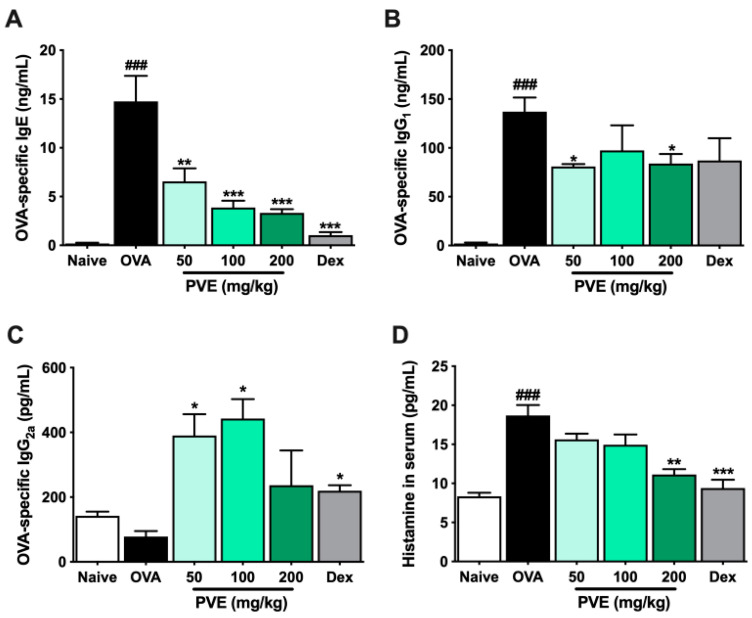
PVE reduced the levels of OVA-specific antibodies and histamine in the serum of the OVA-induced asthmatic mouse model. (**A**) OVA-specific IgE. (**B**) OVA-specific IgG_2a_. (**C**) OVA-specific IgG_1_. (**D**) Histamine in serum. Oral administration of PVE 200 mg/kg significantly down-regulated the levels of OVA-specific IgE, IgG_1_, and histamine in the serum. PVE also strongly up-regulated the level of OVA-specific IgG_2a_ in the serum of asthma mice. * compared to OVA. ###, *** *p* < 0.001; ** *p* < 0.01; * *p* < 0.05. BALF = Bronchoalveolar lavage fluid; Dex = Dexamethasone; Ig = Immunoglobulin; PVE = Phaeanthus vietnamensis extract; OVA = Ovalbumin.

**Figure 7 antioxidants-12-01301-f007:**
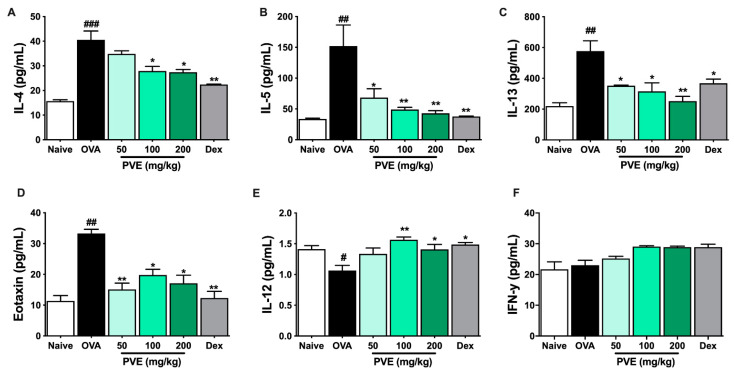
PVE notably increased Th1-related cytokines while decreasing Th2-related cytokines in the BALF of the OVA-induced asthmatic mouse model. The levels of Th2-related cytokines (**A**) IL-4, (**B**) IL-5, (**C**) IL-13, (**D**) eotaxin, and Th1-related cytokines (**E**) IL-12, (**F**) IFN-γ in BALF. The levels of IL-12 in BALF were significantly improved by PVE 200 mg/kg. However, PVE did not increase the level of IFN-γ. The levels of Th2 cytokines IL-4, IL-5, IL-13, and eotaxin in BALF were significantly suppressed by PVE 200 mg/kg or Dex 2.5 mg/kg. # compared to Naive; * compared to OVA. ### *p* < 0.001; ##, ** *p* < 0.01; * *p* < 0.05. BALF = Bronchoalveolar lavage fluid; Dex = Dexamethasone; IFN-γ = Interferon-γ; IL = Interleukin; IL = Interleukin; PVE = Phaeanthus vietnamensis extract; OVA = Ovalbumin; Th = T helper.

**Figure 8 antioxidants-12-01301-f008:**
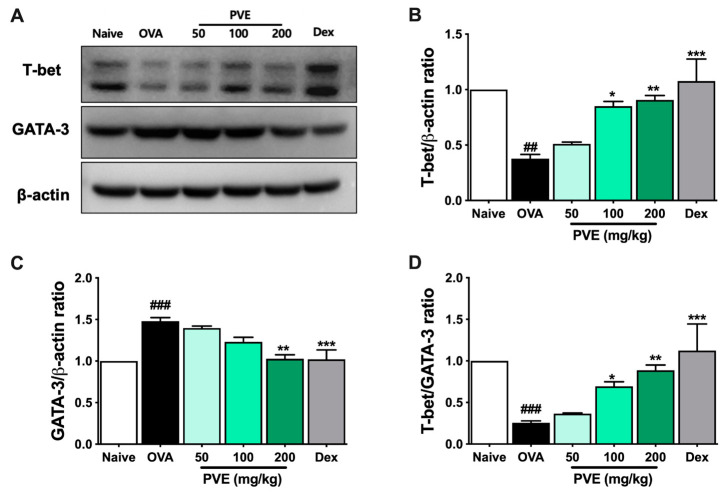
PVE enhanced the expression of Th1-transcription factor T-bet and decreased the Th2-transcription factor GATA-3 in BALF of the OVA-induced asthmatic mouse model. (**A**) Western blot data. The relative expression of (**B**) T-bet and (**C**) GATA-3 in the lung tissue. (**D**) The T-bet/GATA-3 expression ratio. The treatment of Dex or PVE at high dose of 200 mg/kg increased the expression of T-bet and suppressed the expression of GATA-3 in the lung tissue of asthmatic mice. It led to enhancing the T-bet/GATA-3 ratio, which is represented by Th1/Th2 ratio. * compared to OVA. ###, *** *p* < 0.001; ##, ** *p* < 0.01; * *p* < 0.05. BALF = Bronchoalveolar lavage fluid; Dex = Dexamethasone; GATA-3 = GATA Binding Protein 3; IL = Interleukin; PVE = Phaeanthus vietnamensis extract; OVA = Ovalbumin; Th = T helper; T-bet = T-box expressed in T cells.

**Figure 9 antioxidants-12-01301-f009:**
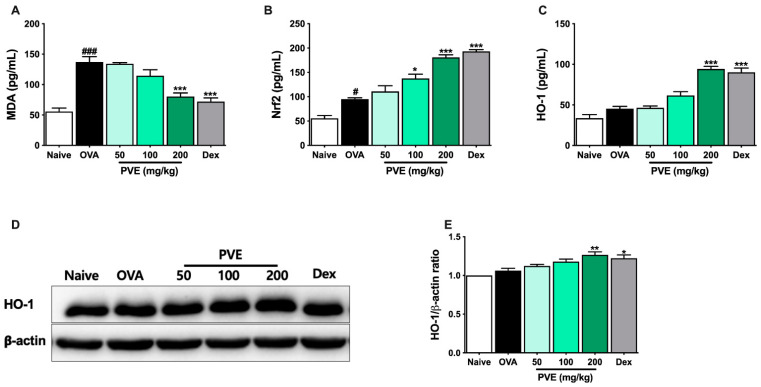
PVE effectively regulated oxidative stress in the OVA-induced asthmatic mouse model. The levels of (**A**) MDA, (**B**) Nrf-2, and (**C**) HO-1 in BALF. (**D**) The expression of HO-1 in the lung tissue. (**E**) The relative level of HO-1 in the lung tissue. # Compared to Naive; * compared to OVA. ###, *** *p* < 0.001; ** *p* < 0.01; #, * *p* < 0.05. BALF = Bronchoalveolar lavage fluid; Dex = Dexamethasone; HO-1 = heme oxygenase-1; MDA = Malondialdehyde; Nrf2 = nuclear factor erythroid 2–related factor 2; PVE = *Phaeanthus vietnamensis* extract; OVA = Ovalbumin.

**Figure 10 antioxidants-12-01301-f010:**
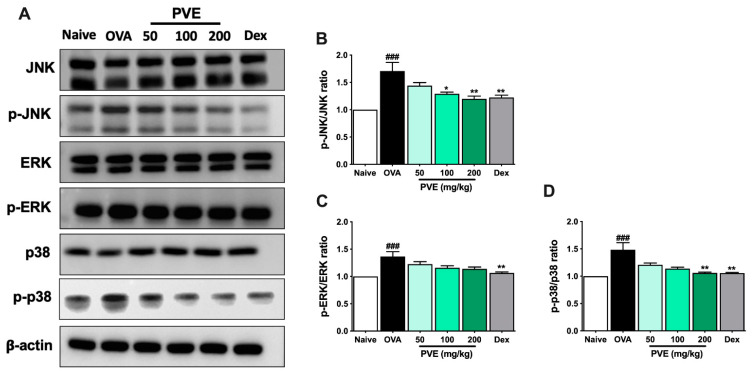
PVE suppressed the MAPK signaling in the OVA-induced asthmatic mouse model. (**A**) Western blot results of MAPKs related protein. The relative levels of (**B**) p-ERK/ERK. (**C**) p-JNK/JNK, and (**D**) p-P38/P-38. * compared to OVA. ### *p* < 0.001; ** *p* < 0.01; * *p* < 0.05. BALF = Bronchoalveolar lavage fluid; Dex = Dexamethasone; MAPK = mitogen-activated protein kinases; PVE = *Phaeanthus vietnamensis* extract; OVA = Ovalbumin.

## Data Availability

The data presented in this study are available on request from the corresponding author.

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
