# Peer review of "Phaeanthus vietnamensis* Ban Ameliorates Lower Airway Inflammation in Experimental Asthmatic Mouse Model via Nrf2/HO-1 and MAPK Signaling Pathway"

_antioxidants, 2023, doi:10.3390/antiox12061301_

Round 1

Reviewer 1 Report

The authors investigate a medicinal plant extract, P.vietnamensis Ban (PVE) and its ability to suppress airway inflammation in an allergic mouse model. This is a proof-of-concept study that shows that a high concentration of PVE is able to suppress mast cell degranulation, some inflammatory cell infiltration into the lung, local inflammation as well as some systemic inflammation.

Specific comments/questions:

Introduction:

·      - I wouldn’t say that the T-helper response, eosinophils and mast cells are “associated” with asthma, they are some of the fundamental driving forces of asthma pathology.

·          - Line 77-80; I’m not sure what the author is trying to say here.

·         -  Refs: I couldn’t locate reference 15, are there any other references the authors could use?

·        -   Intro needs more information in general e.g what are the clinical manifestations and complications of asthma, what are the side effects of corticosteroids? Authors need to provide a compelling reason why alternative therapies are necessary.  

·     - Authors list a number of components of PVE, which they say have anti-inflammatory properties. Could they provide a bit more details about these chemicals? Where have they been used, what exactly can they do?

Methods:

·          - More detail is needed for the PVE preparation, enough so that it can be repeated by others. E.g the 5um cartridge filters, what do they do, where are they from? What is the final concentration of PVE extracted at the end of the process?

·          - Animal protocol – how many mice used/group? Authors say the expt was repeated 3 times, but how many mice/run? PVE is administered orally, but authors don’t say how specifically. I assume by gavage, but what volume? They don’t mention ‘gavage’ until the very last conclusion, but oral could include in the drinking water.

·         -  Importantly, authors call the control group “naïve” mice. This is not an accurate term, naïve suggests that mice received no treatments, but they received saline. This group should be called “Saline”. Also, authors don’t say if these mice received alum with the saline or not.

·          - Need more details for the RPMCs degranulation assay.

·        - Were mice BALFed just once? Its quite difficult to detect cytokines in mouse BALF.

Results:

·          - The legend of Figure 1 says mice were 5 wks old, but methods states 6 wks. Please change accordingly.

·          - Figure 3 legend says “RPMCs were pre-treated with PVE (10, 1, 0.1 mg/ml)” however the figure doesn’t show the 10mg/ml concentration.

·           - Section 3.4 discusses the results for figure 4B before the results for Figure 4A (line 261-266), this should be the other way around.

·           -  Figure 4: authors should list the treatments in the key in the same order as the bars of data in the graph.

·           -  There are a couple of spelling mistakes in section 3.7

·           - Figure 8: Why are there no error bars for the “naïve” data in the graphs?

·          - Section 3.8: Line 366-367 mentions SOD data measured by ELISA, but figure 9 doesn’t show any SOD data.

·          - Line 367 should say “figure 9” not “8”

·     - Line 369 the authors state that “Nrf2 and HO-1 had been notably suppressed compared to naïve mice”. This is not the case, the opposite is true, Nrf2 is increased compared to naïve mice.

·    - Figure 9E – the changes in HO-1/B-actin look extremely small, and barely significant. What statistics did the authors use to show significance between PVE200 and OVA.

Discussion:

·          - Line 425: “epithelial barrier was disrupted then epithelial cells fell into the BALF” – I’m not sure what the authors are trying to say here. Do they mean that the action of BALF-ing dislodged the damaged epithelial cells?

·          - The main issue is that authors suggest that PVE could be a good option for asthma treatment. However, the treatment model used in the study is a proof-of-concept model, in order to suggest that PVE could be used as a treatment, authors would have to use PVE after the establishment of disease (i.e.after day 29 of the model) and show that PVE can still reduce disease features. At best, this study demonstrates that PVE can prevent the development of asthma, but not treat it.  

·         There are a lot of grammatical errors throughout the article, and authors may need help with editing and fixing of those errors.

·         There are a few instances where the words used are not the appropriate word or phrase e.g. in the discussion line 404 “retrieving lung histology”, line 407: “relieving the activation of MAPK”, line 413: “breaking the Th1/Th2 homeostasis”. So authors will need help with using the appropriate words.

Author Response

Thank you very much for your important comments. Please find our reply attached

Reviewer 2 Report

In this study, the authors investigated the potential of Phaeanthus Vietnamensis Ban extracts to alleviate lower airway inflammation in an experimental asthmatic mouse model through the Nrf2/HO-1 and MAPK signaling pathways. The findings of this study are significant as they suggest that PVE could serve as an effective complementary treatment for asthma. The study was well-conducted and effectively presented. Here are my comments:

(1)   The results presented in sections 3.2 and 3.3 are based on in vitro studies, which are preliminary investigations conducted prior to the animal study. Therefore, it may not be suitable to include these results in the main text of the paper. If the authors intended to assess the toxic effects of PVE, they should consider measuring liver and kidney toxic indicators in the serum or urine, or utilizing histological methods. Assessing the cytotoxicity of PVE in vitro alone does not provide comprehensive toxicological information, as the metabolic processes in vivo may differ significantly from those in vitro. Hence, I recommend deleting or relocating sections 3.2 and 3.3 to the supplementary materials.

(2)   The abbreviation "BALF" should be defined with its full English name when first mentioned in the abstract.

(3)   The species name "Phaeanthus Vietnamensis" or "P. Vietnamensis" should be italicized throughout the text. Please make the necessary corrections accordingly.

Author Response

(The authors gave the same response as above.)

Round 2

Reviewer 1 Report

 - Methods - Authors need to add the sentence they provided in their response about mice being gavaged with 200ul (not 200ml) into the manuscript. 

 - Results - section 3.7, line 350, should read PVE-treated not PNE-treated.

 - Conclusion - line 515, should read "prevention". I still don't agree that the authors have shown treatment potential of PVE, unless authors have data to show that on day 16 when they start PVE treatment, mice have all the systemic and lung features of asthma/allergic airways disease. From my experience, mice need to receive the aerosol challenge before exhibiting any lung features of disease. On day 16 of the model mice will only have systemic features of disease, therefore it can't be classified as a treatment if you don't have all the features of disease already present when you start treatment. True treatment would mean starting PVE treatment from day 30 of their model. Authors can say that PVE prevents the progression of disease. 

Author Response

Thank you very much for your comments. Please find the attachment
